# How do associations between sleep duration and metabolic health differ with age in the UK general population?

**Anmol Arora**[1], **David Pell**[2], **Esther M. F. van Sluijs**[2], **Eleanor M. Winpenny**[2]*

**1** School of Clinical Medicine, University of Cambridge, Cambridge, United Kingdom, **2** Centre for Diet and Activity Research (CEDAR) & MRC Epidemiology Unit, University of Cambridge School of Clinical Medicine, Institute of Metabolic Science, Cambridge Biomedical Campus, Cambridge, United Kingdom

* ew470@cam.ac.uk

## Abstract

### Background

Despite a growing body of evidence suggesting that short sleep duration may be linked to adverse metabolic outcomes, how these associations differ between age groups remains unclear. We use eight years of data from the UK National Diet and Nutritional Survey (NDNS) (2008–2016) to analyse cross-sectional relationships between sleep duration and metabolic risk in participants aged 11–70 years.

### Methods

Participants (n = 2008) who provided both metabolic risk and sleep duration data were included. Self-reported sleep duration was standardised by age, to account for differences in age-related sleep requirements. A standardised metabolic risk score was constructed, comprising: waist circumference, blood pressure, serum triglycerides, serum high-density lipoprotein cholesterol, and fasting plasma glucose. Regression models were constructed across four age groups from adolescents to older adults.

### Results

Overall, decreased sleep duration (hrs) was associated with an increased metabolic risk (standard deviations) with significant quadratic (B:0.028 [95%CI: 0.007, 0.050]) and linear (B:-0.061 [95%CI: -0.111, -0.011]) sleep duration coefficients. When separated by age group, stronger associations were seen among mid-aged adults (36-50y) (quadratic coefficient: 0.038 [95%CI: 0.002, 0.074]) compared to other age groups (e.g. adolescents (11-18y), quadratic coefficient: -0.009 [95%CI: -0.042, 0.025]). An increased difference between weekend and weekday sleep was only associated with increased metabolic risk in adults aged 51–70 years (B:0.18 [95%CI: 0.005, 0.348]).

**Data Availability Statement:** The data underlying the results presented in the study are available from the UK Data Service under the 'National Diet

and Nutrition Survey' (https://beta.ukdataservice.ac.uk/datacatalogue/series/series?id=2000033).

**Funding:** The NDNS RP is funded by Public Health England (PHE), an executive agency of the Department of Health, and the UK Food Standards Agency (FSA). This study was supported by the Centre for Diet and Activity Research (CEDAR), a UKCRC Public Health Research Centre of Excellence (https://www.cedar.iph.cam.ac.uk/). Funding from the British Heart Foundation (https://www.bhf.org.uk/), Department of Health (https://www.gov.uk/government/organisations/department-of-health-and-social-care), Economic and Social Research Council (https://esrc.ukri.org/), Medical Research Council (https://mrc.ukri.org/), and the Wellcome Trust (https://wellcome.ac.uk/), under the auspices of the UK Clinical Research Collaboration, is gratefully acknowledged (087636/Z/08/Z; ES/G007462/1; MR/K023187/1; RES-590-28-0002). EvS and EW are supported by the Medical Research Council (MC_UU_12015/7). EW is funded by a fellowship from the Medical Research Council (MR/T010576/1). The funders had no role in study design, data collection and analysis, decision to publish, or preparation of the manuscript.

**Competing interests:** The authors have declared that no competing interests exist.

## Conclusions

Our results indicate that sleep duration is linked to adverse metabolic risk and suggest heterogeneity between age groups. Longitudinal studies with larger sample sizes are required to explore long-term effects of abnormal sleep and potential remedial benefits.

## Introduction

Cardiovascular disease and diabetes remain major public health concerns, with global burdens expected to increase further in the next 10 years. Globally, the economic burden is estimated to rise to at least $2.1 trillion for diabetes and to $1 trillion for cardiovascular diseases by 2030 [1, 2]. Indeed, the World Health Organisation has highlighted cardiovascular disease as the leading cause of death worldwide and diabetes as another common cause of death [3]. Diet and exercise are noted as two key risk factors for the development of metabolic diseases such as cardiovascular disease and type 2 diabetes [4]. However, there have been suggestions that other factors may play a modifiable role in the disease aetiologies, ranging from the gut microbiome to obesogenic HIV drugs [5]. One factor that may contribute to the development of metabolic diseases is short sleep duration. Prior research has found associations between sleep duration and both metabolic outcomes and diet quality in adults [6, 7]. As an example, Potter et al., (2017) noted that, in cross-sectional data from UK adults, short sleep duration was associated with increased body mass index (BMI) and several other metabolic markers [8]. Furthermore, experimental studies in adults have suggested that a relationship between sleep duration and metabolic outcomes may be mediated by increased dietary energy intake as well as physical activity [9]. These findings are supported by a systematic review and meta-analysis [10]. However, there have been relatively few studies analysing the impact of sleep on metabolic outcomes in younger age groups or how these relationships may change with age. Early research with relatively small sample sizes has indicated that sleep may influence metabolic health in these age groups, for example by altering insulin sensitivity [11, 12]. Sleep duration requirements are known to differ with age and, similarly, metabolic health is recognised as varying with age [13, 14]. Adolescents are generally recommended 8 to 10 hours sleep per night, decreasing with age to 7 to 8 hours sleep for adults aged over 65 [15–19]. The potentially complex interplay between age, sleep architecture and metabolic health is an understudied area of research. It has been suggested that factors such as diet may exert heterogeneous effects depending on age and this could also be true of sleep health [20].

Two key metrics in describing sleep are quality and duration. Early research has suggested the existence of associations between sleep duration and diet in young people [21, 22]. One of these studies even noted that sleep restriction in childhood was associated with less favourable BMI profiles in older age [23]. The authors concluded that addressing unhealthy sleep patterns in childhood may offer some relief to the increasingly burdensome obesity crisis. However, this remains a contestable area of research with others finding no such relationship in adolescents [24, 25]. Similarly, according to a recent review, the evidence between fluctuations in sleep duration between weekday and weekend nights and metabolic risk in young people is limited and inconclusive [26]. There is, however, a growing body of literature indicating that the difference between weekend and weekday night sleep is associated with unhealthy changes to eating behaviours and diet quality, though this is heavily reliant on cross-sectional data [27, 28]. Eating behaviours and diet quality would likely lie on a causative pathway between sleep and metabolic risk. In order to develop appropriate recommendations, there is a need for

better understanding of how sleep affects metabolic risk amongst different age groups, which may call for heterogeneous interventions.

In this study, we focus on sleep duration, which is both modifiable and relatively easy to record [29]. Sleep duration can be self-reported or measured using medical devices including polysomnograms [30]. Although device-measured sleep duration is more precise, self-reported sleep duration is more feasible in large-scale epidemiological research, due to lower cost and expertise requirements [31]. For the purposes of this study, we use self-reported sleep duration as our exposure variable. There are a number of risk factors recognised to increase the risk of developing heart disease, type 2 diabetes and strokes. Central obesity, insulin resistance, hypertension and dyslipidaemia are particularly important and together they are defined as metabolic syndrome [32]. In our analysis, we use a standardised metabolic risk score as an outcome variable which incorporates these risk factors in order to ensure that our outcome variable truly corresponds to risk of future health outcomes, including cardiovascular disease [33].

In this study, we use data from years 1–8 of the UK-representative 'National Diet and Nutrition Study Rolling Programme (NDNS)' (2008–2016) to compare associations between sleep duration and metabolic risk score in different age groups. Specifically, we aim to address the research question: 'How do associations between sleep duration and metabolic health differ with age in the UK general population?'

## Methods

### Survey design and participants

The NDNS is an annual cross-sectional survey which assesses the diet of the general population of the UK and includes measures on health. The NDNS aims to recruit 1000 participants each year, comprising an equal ratio of adults (aged 19 years and older) and children (aged 1.5 to 18 years). Households were sampled from the U.K. Postcode Address File, a list of all addresses in the U.K., with up to one adult and one child from each household eligible for inclusion in the survey [34]. The survey consisted of two relevant stages, the first of which was a computer-assisted interview and the second was a nurse visit in which blood samples and waist circumference measurements were taken. Only a smaller proportion consent to the nurse visit stage of the study, during which participants provide metabolic data [35]. Written informed consent was obtained from all participants, or from their parents/guardians if aged under 16 years. Ethical approval for the NDNS was obtained from the Oxfordshire A Research Ethics Committee and the Cambridge South NRES Committee (Ref. No. 13/EE/0016). In this analysis, we use data on participants aged from 11 to 70 years, combined from the first eight years (2008–2016) of the NDNS to provide a sufficiently large sample size for age-based sub-population analysis (Table 1) [36]. Metabolic data was unavailable to be analysed for participants aged below 11 years. An upper age limit of 70 years was set due to the likelihood of unrecorded comorbidities affecting sleep in the elderly population. We excluded participants from the analyses if they reported taking anti-hypertensive or lipid-lowering medication.

### Sleep duration

In the survey, sleep data were reported in stage 1 (the interviewer stage) of the data collection, in response to two questions which captured both weekend and weekday night sleep duration: '*Over the last seven days, that is since (date), how long did you usually sleep for on week nights. That is Sunday to Thursday nights?*' and '*And over the last seven days, how long did you usually sleep for on weekend nights. That is Friday and Saturday nights?*'. We calculated sleep duration as the average self-reported quantity of sleep obtained by respondents of the NDNS over the seven days prior to questioning, over both weekday and weekend nights.

**Table 1. Descriptive statistics of the weighted sample, grouped by age category, from the NDNS Years 1–8.**

| | | Age category | | | | |
|---|---|---|---|---|---|---|
| | | All ages | Adolescents | Young adults | Mid-aged adults | Older adults |
| | | (n = 2008) | 11–18 | 19–35 | 36–50 | 51–70 |
| | | | (n = 431) | (n = 423) | (n = 632) | (n = 522) |
| Sex (%) | Male | 47.3 | 50.7 | 48.5 | 47.4 | 44.6 |
| | Female | 52.7 | 49.3 | 51.5 | 52.6 | 55.4 |
| National Statistics socio-economic classification (NS-SEC3) of household reference person (%) | Managerial and professional occupations | 50.0 | 41.9 | 42.2 | 56.2 | 54.5 |
| | Intermediate occupations | 19.8 | 19.9 | 21.3 | 17.5 | 20.9 |
| | Routine and manual occupations | 28.1 | 35.1 | 34.5 | 23.8 | 23.4 |
| | Never worked and other | 2.1 | 3.1 | 2.0 | 2.5 | 1.2 |
| Ethnic group (%) | White | 88.5 | 88.0 | 83.6 | 87.2 | 96.4 |
| | Mixed ethnic group | 2.4 | 3.1 | 4.1 | 1.3 | 1.3 |
| | Black or Black British | 2.5 | 2.1 | 3.9 | 2.8 | 0.7 |
| | Asian or Asian British | 5.2 | 5.0 | 7.2 | 6.4 | 1.3 |
| | Any other group | 1.4 | 1.8 | 1.2 | 2.2 | 0.3 |
| Smoking status (%) | Never a regular smoker | 65.0 | 90.0 | 62.7 | 61.0 | 61.1 |
| | Ex-smoker | 20.4 | 0.7 | 16.5 | 23.6 | 30.2 |
| | Current smoker | 14.6 | 9.4 | 20.8 | 15.3 | 8.7 |
| Screen time (hours/day) (s.d.) | Mean screen time at home per day (computer use, TV, DVD or video viewing) | 5.0 (2.2) | 5.9 (2.2) | 5.3 (2.5) | 4.7 (2.1) | 4.8 (1.9) |
| Sleep duration (hours/day) (s.d.) | Average sleep duration in seven days prior to questioning | 7.4 (1.2) | 8.6 (1.3) | 7.6 (1.1) | 7.1 (1.2) | 7.0 (1.1) |
| | Average weeknight sleep (Sunday to Thursday) in seven days prior to questioning | 7.3 (1.3) | 8.4 (1.5) | 7.5 (1.1) | 7.0 (1.2) | 6.9 (1.2) |
| | Average weekday sleep (Friday to Saturday) in seven days prior to questioning | 7.7 (1.5) | 9.1 (1.8) | 7.9 (1.5) | 7.5 (1.4) | 7.2 (1.2) |
| Difference between weekend and weekday night sleep (hours) (s.d.) | Mean weekend sleep minus mean weekday night sleep (Note: positive value indicates greater sleep at the weekend) | 0.4 (1.2) | 0.7 (2.0) | 0.4 (1.3) | 0.5 (1.0) | 0.3 (0.8) |
| Metabolic outcome | Mean standardised metabolic risk score (MetZscore) (s.d.) | -0.027 (1.02) | 0.021 (0.75) | 0.035 (0.88) | -0.003 (1.15) | -0.153 (1.06) |
| | Waist circumference (cm) | 88.7 (13.9) | 76.0 (10.7) | 85.9 (13.6) | 91.9 (13.1) | 92.8 (12.8) |
| | Triglycerides (mmol/L) | 1.2 (0.8) | 0.9 (0.5) | 1.1 (0.7) | 1.4 (1.1) | 1.3 (0.7) |
| | HDL (mmol/L) | 1.5 (0.4) | 1.4 (0.3) | 1.4 (0.4) | 1.4 (0.4) | 1.6 (0.5) |
| | Glucose (mmol/L) | 5.1 (1.0) | 4.8 (0.5) | 4.9 (0.9) | 5.2 (1.2) | 5.3 (1.0) |
| | Average blood pressure (mm/Hg) | 97.2 (12.2) | 88.3 (8.2) | 93.8 (10.1) | 99.0 (12.3) | 102.4 (12.9) |

To generate an age-adjusted measure of sleep duration that could be compared between participants of different ages, we regressed sleep duration against age to produce a residual sleep duration value for each participant. Since change in sleep with age is non-linear, a restricted cubic spline regression was used, with 5 knots at ages 14, 26, 39, 50 and 65 years, to ensure appropriate adjustment for age-related variation. The resultant 'age-adjusted sleep' measure indicates sleep duration in hours relative to the average value for that age. We also subtracted the mean weekday night sleep duration from mean weekend sleep duration for each participant to calculate a measure of difference in sleep duration (in hours) between weekday and weekend nights.

## Metabolic risk factors

To provide an overall measure of metabolic risk, we used metabolic risk z-scores (MetZscore) [37–39]. Use of a continuous risk score, standardised for age, facilitates comparison across age groups [40]. The MetZscore was based on the metabolic risk factors included in the definition of metabolic syndrome from the National Cholesterol Education Program—Adult Treatment Program III (NCEP-ATPIII): waist circumference, blood pressure (BP), serum triglycerides, serum high-density lipoprotein (HDL) cholesterol, and fasting plasma glucose [41]. Metabolic risk factors were measured during stage 2 (the nurse visit) of the data collection. Waist circumference was measured at the midpoint between the iliac crest and lower rib, using a tape measure precise to the nearest 0.1cm. Systolic and diastolic BP values were measured in a seated position ensuring that participants had not eaten, consumed alcohol, exercised, or smoked in the preceding 30 minutes. Three BP readings were taken at one-minute intervals. Serum triglycerides, serum HDL cholesterol and plasma glucose were measured from fasted blood samples, Details of each assay and their quality control data are provided in NDNS Appendix Q [35].

The MetZscore was calculated, as has been described previously, by first standardising the individual metabolic risk factors by regressing them onto selected demographic variables (age, sex, ethnicity) [20, 37]. To account for the non-linearity of associations between age and metabolic outcomes, a restricted cubic spline regression was applied to the weighted sample for each outcome, with knots at ages 15, 20, 30, 40 and 50 years, to ensure appropriate adjustment for age-related variation. The resulting residuals were standardised to give a mean of 0 and a standard deviation (SD) of 1. Standardised HDL is inversely related to metabolic risk so it was multiplied by -1. The residuals of systolic and diastolic BP were averaged to give a value of mid-blood pressure. Z-scores for the individual risk factors were summed to create the MetZscore. This score was standardised to have a mean of 0, with a standard deviation of 1, within the analysis population. A higher MetZscore is indicative of a less favourable metabolic syndrome profile.

## Covariates

Age, sex, ethnicity, smoking and screen time were self-reported. Ethnicity was classified according to five groups (white, mixed, black, Asian and other). Smoking was categorised into three categories: 'current smoker', 'ex-regular smoker' and 'never regular smoker'. Television viewing time has been used as a covariate in comparable studies; the NDNS provided data on television viewing together with other related activities to produce a measure of screen time [23]. Screen time was reported in response to a series of questions on how much time participants spent watching DVDs, TV, or videos and computer use, before 6pm and after 6pm on both weekends and weekday nights. These data were summed to create a single variable of total weekly screen time. National statistics socio-economic classification (NSSEC) of the household reference person was reported by the household reference person (defined as the household member with the highest income), and classified into four groups: Managerial and professional occupations; Intermediate occupations; Routine and manual occupations; and Never worked or other [42]. We did not adjust for diet quality or physical activity as the literature suggests that these are likely to be on the causal pathway between sleep and metabolic outcomes; this was not the case for screen time [9]. We considered inclusion of marital status, alcohol intake and employment status as covariates, but these were not associated with the exposure and outcome variables in our dataset.

## Statistical analysis

All the analyses were performed using R (version 3.6.0). The weights (blood sample weights) provided with the NDNS dataset were applied to account for clustering and response biases. In our descriptive analysis, sociodemographic factors, sleep and MetZscore were summarized overall and by age category. 562 participants had missing screen time data, 40 had missing socioeconomic classification data and 13 had missing data regarding smoking habits. Multiple imputations by chained equations were used to impute missing data for covariates, under the missing at random assumption, following recommendations from White et al. [43]. Twenty imputed datasets were generated using the R command 'mice()'. Each dataset was analysed separately and results combined using Rubin's rule [43].

We first tested associations between age-adjusted sleep and MetZscore across all age groups combined. We compared linear and quadratic models, by calculating residual sum of squares and Akaike information criterion (AIC), both of which indicated that the quadratic model provided a better fit to the data. Four age groups were constructed: 11-18y, 19-35y, 36-50y and 51-70y and 'age group' was included as an interaction term in the regression models to test differences in association by age. Groups were designed such that they included a comparable number of participants whilst still separating adolescents from adults, due to their different sleep duration requirements. Covariates were added in three stages: the first included no covariates to produce an unadjusted model (model 1). Model 2 included sex, ethnicity and socioeconomic classification and was performed with and without age group as a moderating factor. Finally, model 3 included the previous covariates with the addition of screen time and smoking status.

## Results

### 1. Characteristics of study population

Of 12097 participants who took part in years 1–8 of the NDNS, 7860 participants were aged between 11 and 70 years. Of these, 2627 had complete metabolic data and 2365 had complete sleep and metabolic risk data. A further 357 were excluded due to taking anti-lipid or anti-hypertension medication. This left 2008 individuals in the analysis, with 21.5% aged 11 to 18y, 21.1% aged 19 to 35y, 31.5% aged 36 to 50y and 26.0% aged 51 to 70y. Descriptive data on the study population are shown in Table 1.

The data show an average sleep duration of 7.4 hours per night overall, with more sleep recorded in adolescents. Fig 1 illustrates the relationship between age and sleep duration. Sleep duration decreases steeply with age through adolescence then slows but continues to decrease as age increases.

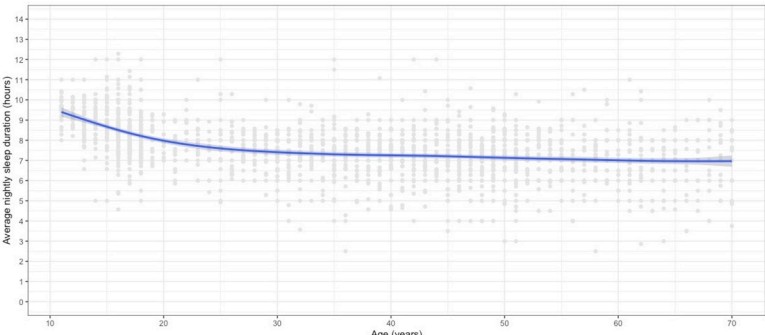

**Fig 1. The unadjusted relationship between sleep and age, across all age groups (95% CI shown in grey).**

## 2. Relationship between sleep duration and metabolic outcomes

Overall, the results in Table 2 indicate that there is an association between increased sleep (in hours) and reduced metabolic risk (in standard deviations) across all ages (quadratic term: 0.028 [95%CI: 0.007, 0.050, p = 0.01], linear term: -0.061 [95%CI: -0.111, -0.011, p = 0.02]). When age categories are introduced, the association appears only in the 36–50 age group, which suggests that the overall effect may be driven by the effect seen in this group. The coefficients presented in Table 2 refer to regression equation: $y = ax^2 + bx + cz + \epsilon$, where y represents change in metabolic risk score, x represents age-adjusted sleep duration, z represents one of the covariate terms and $\epsilon$ represents the error term.

The positive quadratic term (0.028 [95%CI: 0.007, 0.050, p = 0.01]) in the results for all ages combined indicate that as sleep duration increases away from the mean, its marginal effect on metabolic risk increases. Therefore, the greater the reduction in sleep time, the larger the corresponding increase in metabolic risk, as illustrated in Fig 2. Fig 2 also suggests that sleep durations above the mean may be associated with increasing metabolic risk as sleep duration increases; however, this effect is not significant.

When we compare the associations between different age groups, we see that for mid-aged adults the coefficient of the quadratic sleep duration term (0.038 [95%CI: 0.002, 0.074, p = 0.04]) is much larger than that seen in adolescents (-0.009 [95%CI: -0.042, 0.025, p = 0.59]), suggesting that associations of metabolic risk with low sleep duration are less marked in a younger population (Fig 3).

## 3. Relationship between intra-week sleep fluctuations and metabolic outcomes

As shown in Table 3, the difference between weekday and weekend nightly sleep only appears to be statistically significantly associated with metabolic outcomes in older adults, where it is associated with increased metabolic risk. For adults aged between 51 and 70y, we see that

**Table 2. Cross-sectional associations between sleep duration and metabolic risk score, presenting coefficients for both the quadratic sleep duration term and the linear sleep duration term from the regression model.**

| | Change in metabolic risk score (standard deviations) for sleep duration increase of one hour [95% CI] | | | | | |
|---|---|---|---|---|---|---|
| | Model 1, unadjusted | | Model 2, adjusted using basic covariates (sex, ethnicity, socioeconomic classification) | | Model 3, covariates: sex, ethnicity, socioeconomic classification, smoking, screen time | |
| | Coefficient of age-adjusted quadratic sleep duration term | Coefficient of age-adjusted sleep duration term | Coefficient of age-adjusted quadratic sleep duration term | Coefficient of age-adjusted sleep duration term | Coefficient of age-adjusted quadratic sleep duration term | Coefficient of age-adjusted sleep duration term |
| All ages (n = 2008) | 0.031 [0.010, 0.053]* | -0.067[-0.118, -0.016]* | 0.031 [0.009, 0.053]* | -0.069 [-0.120, -0.019]* | 0.028[0.007, 0.050]* | -0.061 [-0.111, -0.011]* |
| Adolescents 11–18 years (n = 431) | -0.010 [-0.039, 0.020] | 0.052 [-0.020, 0.125] | -0.006 [-0.039, 0.026] | 0.047 [-0.031,0.125] | -0.009 [-0.042, 0.025] | 0.041 [-0.039, 0.121] |
| Young adults 19–35 years (n = 423) | 0.034[-0.010, 0.078] | -0.086 [-0.197, 0.024] | 0.032 [-0.013, 0.077] | -0.078 [-0.187,0.030] | 0.029 [-0.013, 0.072] | -0.058 [-0.161, 0.044] |
| Mid-aged adults 36–50 years (n = 632) | 0.046 [0.009, 0.083]* | -0.091 [-0.191, 0.009] | 0.043[0.006, 0.079]* | -0.096 [-0.193, 0.001] | 0.038[0.002, 0.074]* | -0.092 [-0.189, 0.005] |
| Older adults 51–70 years (n = 522) | 0.019 [-0.017, 0.056] | -0.066 [-0.151, 0.020] | 0.022 [-0.015, 0.060] | -0.070 [-0.156, 0.016] | 0.021 [-0.017, 0.060] | -0.064 [-0.150, 0.021] |

Note

* indicates p<0.05

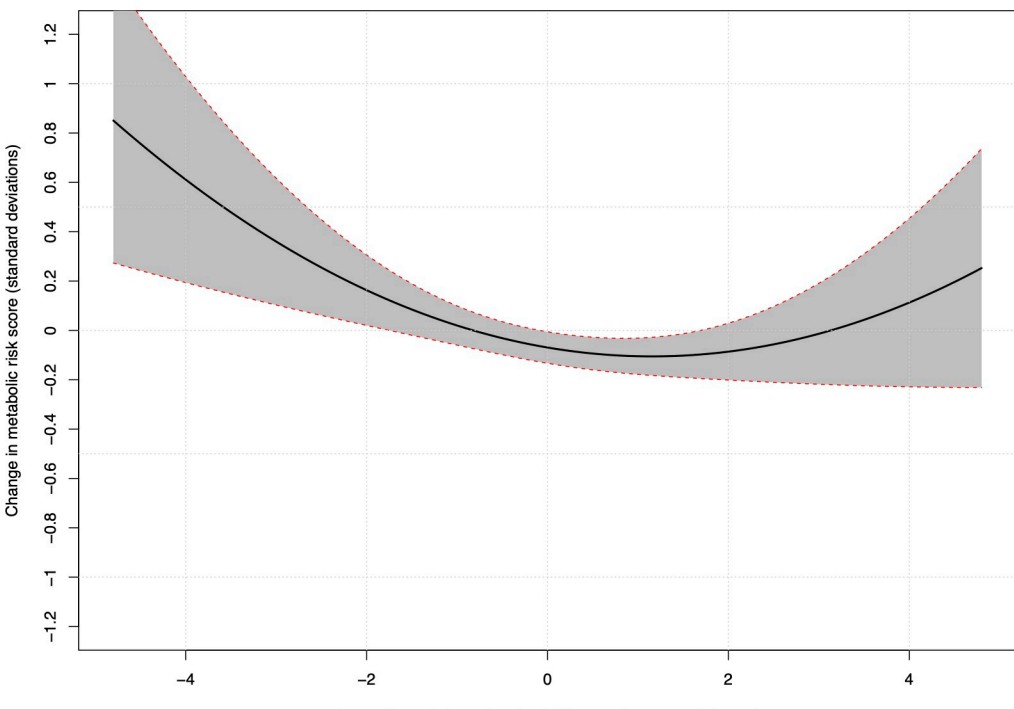

**Fig 2. Model predictions of metabolic risk score.** Model predictions of metabolic risk score (with 95% confidence intervals) as sleep durations vary above and below the mean (model 3, all age groups).

where the difference between weekday (Sunday to Thursday) and weekend nights (Friday to Saturday) increases by one hour, the MetZscore increases by 0.18 SD [95%CI: 0.005, 0.348, p = 0.04]. The standardised score allows for estimation of centiles within a population and a score of 0.18 corresponds to an increase of 7 percentiles.

## Discussion

### 1. Summary of key findings

Our results illustrate that when all age groups are combined, longer sleep duration is associated with more favourable metabolic health. When participants were stratified by age group, we found that this effect was only observed in those aged between 36-50y. In the adolescent age group in particular, the association between sleep and metabolic outcomes appears to be much smaller than in mid-aged adults. It is likely that the mid-aged age group is driving the overall effect. Whilst there may be adverse metabolic outcomes associated with prolonged sleep well above average sleep duration, our findings do not indicate this. The difference between week-end and weekday night sleep duration showed associations with metabolic outcomes only in older age groups, with statistically significantly associations seen only in participants aged 51-70y.

### 2. Strengths and limitations

A major strength of this study was the use of data from the NDNS, a national survey designed to be representative of the UK general population. We used survey weights which are specifically designed to account for non-response to the blood sample collection so that, despite some further participant exclusions specific to this study, our findings can be said to be broadly

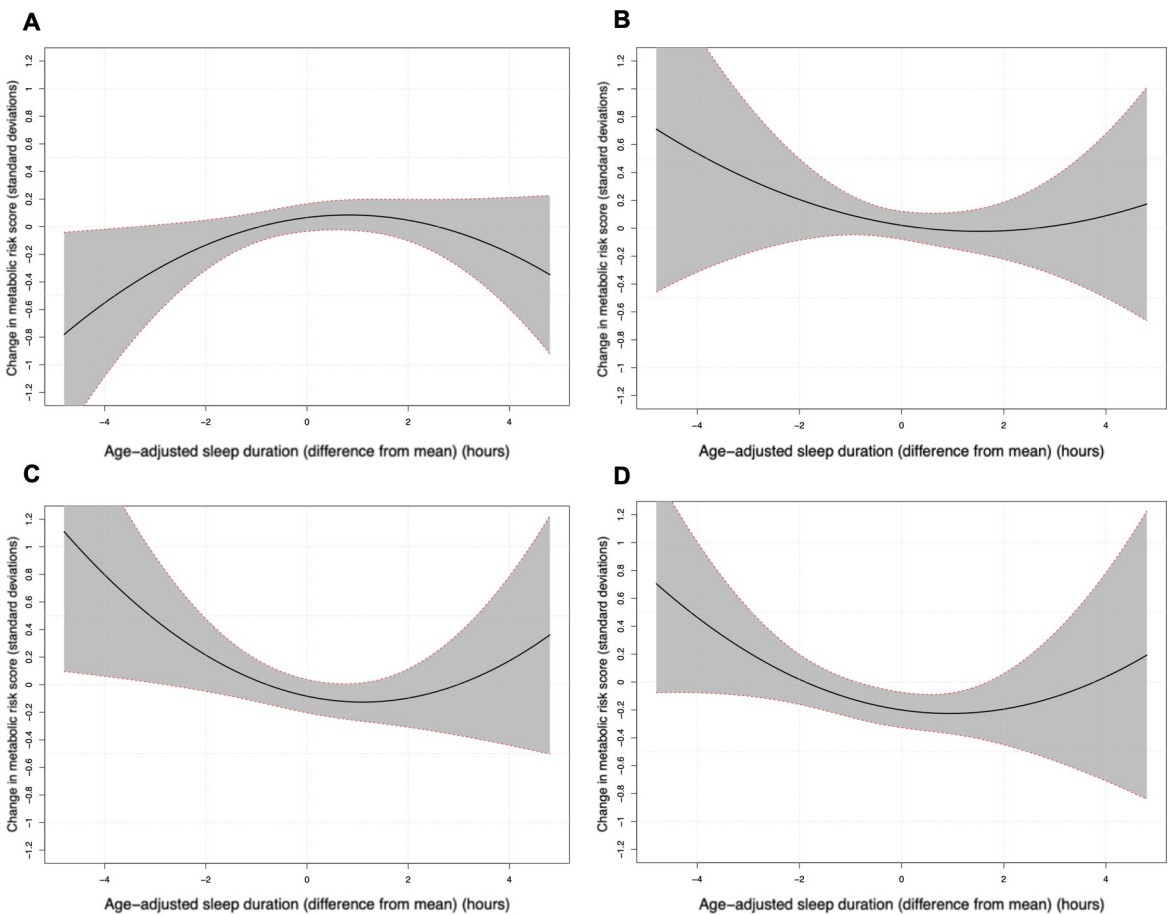

**Fig 3. Model predictions of metabolic risk score by age group.** Model predictions of metabolic risk score (with 95% confidence intervals) as sleep durations vary above and below the mean, separated by age group (model 3, A: age 11 to 18 years, B: 19 to 35 years, C: 36 to 50 years, D: 51 to 70 years).

**Table 3. Cross-sectional associations between the difference between weekend and weekday night sleep and metabolic risk score.**

| | Change in metabolic risk score (s.d.) for sleep duration or sleep duration increase of one hour [95%CI] | | |
|---|---|---|---|
| | **Unadjusted models** | **Adjusted models using basic covariates (sex, ethnicity, socioeconomic classification)** | **Adjusted models (covariates: sex, ethnicity, socioeconomic classification, smoking, screen time)** |
| | **Coefficient of 'difference between weekday and weekend sleep' term** | **Coefficient of 'difference between weekday and weekend sleep' term** | **Coefficient of 'difference between weekday and weekend sleep' term** |
| All ages (n = 2008) | 0.052 [-0.003,0.107] | 0.048 [-0.008, 0.104] | 0.051 [-0.007, 0.108] |
| Adolescents 11–18 years (n = 431) | 0.026 [-0.020, 0.073] | 0.021 [-0.030, 0.071] | 0.015 [-0.037, 0.068] |
| Young adults 19–35 years (n = 423) | 0.001 [-0.097, 0.098] | -0.004 [-0.105, 0.096] | 0.003 [-0.101, 0.107] |
| Mid-aged adults 36–50 years (n = 632) | 0.096 [-0.043, 0.236] | 0.105 [-0.034, 0.243] | 0.104 [-0.033, 0.241] |
| Older adults 51–70 years (n = 522) | 0.176 [0.004, 0.349]* | 0.160 [-0.010, 0.330] | 0.176 [0.005, 0.348]* |

Note

* indicates p<0.05

reflective of the UK population across our included age range. However, the fact that the study population, due to the ethnicity of the UK population, was mostly of white ethnicity is a limitation and complicates the generalisability of the findings. The wide range of data included in the NDNS survey, allow us to test, and where appropriate adjust for, all putative covariates. We were able to exclude participants who were taking blood-pressure or lipid-lowering medications from the analysis, as this would have affected their metabolic risk profile, but unfortunately data was unavailable to allow us to exclude participants taking medication for sleep-related reasons. The NDNS has a well-defined methodology which is consistent between years of the survey, allowing us to combine data across recent years of survey data. Survey data is accompanied with much detail about the interviewer questions and measurement procedures, to ensure that the results may be appropriately compared with those using other data.

Limitations include the self-reported assessment of sleep duration and use of cross-sectional data. Personal experience is highly subjective and previous research has noted that people tend to overestimate sleep duration when self-reporting [44–46]. It has also been noted that psychosocial factors influence sleep reporting and whilst some groups might overestimate their sleep, others may underestimate it [47]. The implication of this is that if the accuracy self-reported sleep duration was associated with certain characteristics this would require correction to mitigate regression dilution bias, though no relevant associations were known at the time of this study. It is possible to measure sleep clinically e.g. by polysomnogram or accelerometer study; however, this was not used in the NDNS. Whilst more precise, the cost and participant burden of polysomnograms reduces their applicability in clinical practice whereas self-reported sleep duration can be routinely assessed in practice. Another limitation to our study is the use of cross-sectional data, which is only capable of elucidating correlation between variables. Whilst the literature generally favours that sleep health is a potential driver of metabolic health, the possibility of reverse causality cannot be ignored.

## 3. Comparison with previous evidence

Our results indicated a decrease in sleep duration with age, consistent with existing literature. It has been noted that people typically suffer from well-characterised changes to sleep architecture as a function of ageing, with reduced sleep duration requirements [14, 48]. Indeed, sleep duration recommendations tend to reflect this with adults recommended less sleep than adolescents and children; some guidelines have now begun to refine sleep duration recommendations for the elderly [15]. In most cases, guidelines recommend around 8 to 10 hours sleep for adolescents and 7 to 9 hours sleep for adults, with some recommending 7 to 8 hours sleep for adults aged over 65 [15–19]. It is worth noting that the recommendations are designed with a high granularity during adolescence and early childhood, such that they change at frequent intervals with age in these age groups [18]. We were unable to draw conclusions on participants aged over 70 years old, due to the small number of eligible participants in the age group as well as the high prevalence of comorbidities in this age group confounding metabolic health measurements. However, it has been noted that, especially in this age group, additional characteristics such as sleep quality have increased importance [14].

Others have reported that sleep durations both above and below recommendations are associated with unfavourable health outcomes [14]. However, the metabolic health effects of excessively prolonged sleep appear inconclusive in the literature. Our results are, overall, consistent with a recent meta-analysis which found that short sleep duration is associated with increased risk of metabolic syndrome and that prolonged sleep duration was not associated with an increased risk [49]. However, at least one other study has suggested a U-shaped relationship between sleep duration and metabolic risk [50]. This is recognised as a conflicting

area of research, with a systematic review noting that in another meta-regression analysis authors found a relationship between long sleep and adverse metabolic outcomes; however, they noted the need for future research exploring potential mechanisms and examining the precise nature of any potential relationship [51]. Future research may also examine any reverse causality and whether metabolic factors exert a causal effect in reducing sleep duration. There is a notable shortage of longitudinal studies exploring the long term effects of sleep deprivation on metabolic health. One prospective cohort study, conducted using New Zealand data, found that short sleep as a child was linked with increased long-term metabolic risk at age 32 [23]. The authors concluded that addressing sleep duration may be a useful tool in stemming the ongoing obesity epidemic. Our results suggest that strengths of associations between sleep duration and metabolic health might be heterogeneous between age groups. The results of the New Zealand study demonstrate long term effects of sleep deprivation in childhood; this is not necessarily inconsistent with our findings, which explored cross-sectional associations. We found that poor sleep was associated with adverse metabolic health effects in mid-aged adults, but it is possible that the individuals driving the effect in this group also slept poorly in child-hood, for these effects to only manifest later in life.

The difference between weekend and weekday nightly sleep has been identified as an important avenue for future studies, with a limited body of literature at present [52]. Our descriptive results agree with research noting that there is reduced intra-week sleep variability in older age groups. One study noted that the difference between weekend and weekday sleep may be a healthy phenomenon to compensate for reduced sleep on work/school nights [52]. It may be that younger age groups are more resilient to short-term fluctuations in sleep duration. Indeed, we see that the youngest age group (11–18 years) exhibits the largest mean difference between weekend and weekday sleep, but showed no association of this difference with metabolic risk score, whereas the oldest age group (51–70 years), where associations were seen, exhibits the lowest. Older participants may be more likely to have flexible work times or be retired and so may not have to wake up earlier on weekdays, compared to school or working age participants. In older age groups, whose nightly sleep is typically less variable, any fluctuations may be more likely to be pathological or deleterious to metabolic health. Notably, some research has found differences between weekend and weekday sleep to be associated with increased hunger, but not with increased fat mass [27]. However, conflicts in the literature are apparent with another study finding that such difference was associated with an array of metabolic risk factors, including adiposity, in adults aged 30–54 years [53].

It has been suggested that changes in circadian rhythm may impact dietary behaviours, with a growing body of evidence, this includes a study noting that adults working night shifts are more likely to choose unhealthy breakfast options than those working day shifts [27, 54, 55]. There is also a well-described relationship between shift work and metabolic risk factors, though there is a need for further longitudinal studies in this area with some studies producing discordant results [56, 57]. Overall, the literature suggests the existence of an understudied, but complex, interplay between circadian rhythm shifts and health outcomes. The difference between weekend and weekday sleep represents just one of several potential variables to denote circadian rhythm variability.

## 4. Implications of findings

Sleep is currently an understated marker of human health at a population level; it is also sel-dom considered in routine clinical practice as a health marker [58]. In order to warrant wide-spread assessment of sleeping habits as a diagnostic marker or potential intervention we must consider whether such application would be clinically useful. For this we need to consider

feasible interventions, which will depend upon the underlying cause. Screening for poor sleep duration may help in the diagnosis and prevention of mental and physical conditions, which could be either a cause or consequence of poor sleep [59]. There are a range of causes of poor sleep quality, some of which include: head trauma, depression, anxiety, arthritis, chronic pain or hyperthyroidism [60, 61]. Identification that a patient is experiencing poor quality of sleep is inadequate to formulate a diagnosis but it may aid in screening for a number of conditions, including, but by no means limited to, insomnia. Indeed, the growing body of evidence describing the effects of poor sleep has prompted calls for sleep to be routinely assessed in clinical practice [62]. There may be scope to ask questions about sleep in the same way that questions about smoking, alcohol intake and lifestyle are asked. This would take seconds during regular health check-ups, yet would enable access to an otherwise under-utilised health marker. However, it should be noted that despite a growing body of evidence indicating that poor sleep is associated with health outcomes, there is a shortage of studies assessing whether improving sleep may have therapeutic, or public health, benefits in patients with unfavourable metabolic risk factors [63].

## Concluding remarks

The data presented in this paper indicate that the relationship between sleep duration and metabolic health exhibits heterogeneity between age groups. This reinforces the need for age-appropriate guidelines when expanding the inclusion of sleep health in health consultations. It also suggests that future research into the complications of poor sleep should not assume that effects would be homogeneous between age groups. There is a growing body of literature suggesting that irregular sleep patterns may act as an independent factor in influencing metabolic outcomes. This remains an active area of research, with several gaps in the literature. Whilst existing literature has separated children and adolescents from adults, our data suggest that further age separation in young adults (19 to 35 years) and older adults may too be important. This study is not able to elucidate a causal relationship between poor sleep and poor metabolic outcomes, which would require longitudinal studies. Further research may include examining other outcomes affected by sleep that may feed into metabolic pathways associated with less favourable metabolic profiles. Technological advancements, including fitness trackers and smart watches capable of monitoring sleep, may soon offer precise measurements of sleep duration and propel future research in the field.

## Author Contributions

**Conceptualization:** Eleanor M. Winpenny.

**Formal analysis:** Anmol Arora.

**Funding acquisition:** Esther M. F. van Sluijs.

**Methodology:** Eleanor M. Winpenny.

**Supervision:** Eleanor M. Winpenny.

**Validation:** David Pell.

**Visualization:** Anmol Arora.

**Writing – original draft:** Anmol Arora, Eleanor M. Winpenny.

**Writing – review & editing:** Anmol Arora, David Pell, Esther M. F. van Sluijs, Eleanor M. Winpenny.

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
