## [Decision Letter · Decision Letter 0]

22 Oct 2020

PONE-D-20-29508

How do associations between sleep duration and metabolic health differ with age in the UK general population?

PLOS ONE

Dear Dr. Winpenny,

Thank you for submitting your manuscript to PLOS ONE. After careful consideration, we feel that it has merit but does not fully meet PLOS ONE’s publication criteria as it currently stands. Therefore, we invite you to submit a revised version of the manuscript that addresses the points raised during the review process.

We look forward to receiving your revised manuscript.

Kind regards,

Antonio Palazón-Bru, PhD

Academic Editor

PLOS ONE

Journal Requirements:

2. Please include additional information regarding the survey or questionnaire used in the study and ensure that you have provided sufficient details that others could replicate the analyses. For instance, if you developed a questionnaire as part of this study and it is not under a copyright more restrictive than CC-BY, please include a copy, in both the original language and English, as Supporting Information, or include a citation if it has been published previously.

Reviewers' comments:

Reviewer's Responses to Questions

**Comments to the Author**

1. Is the manuscript technically sound, and do the data support the conclusions?

Reviewer #1: Yes

Reviewer #2: Yes

2. Has the statistical analysis been performed appropriately and rigorously? 

Reviewer #1: Yes

Reviewer #2: Yes

3. Have the authors made all data underlying the findings in their manuscript fully available?

Reviewer #1: Yes

Reviewer #2: Yes

4. Is the manuscript presented in an intelligible fashion and written in standard English?

Reviewer #1: Yes

Reviewer #2: Yes

5. Review Comments to the Author

Reviewer #1: PONE-D-20-29508 “How do associations between sleep duration and metabolic health differ with age in the UK general population?”

In this manuscript, the authors report findings from a cross-sectional study to test whether short sleep duration is related to increased metabolic risk across four age groups from adolescents to older adults. This study has several strengths, including a large representative sample spanning adolescence to older adulthood, objectively-measured metabolic health, MICE to account for missing covariate data, and consideration of both linear and quadratic relationships between sleep and metabolic health. However, there are a number of areas which warrant further description and justification that reduced enthusiasm for the manuscript in its present form:

Introduction:

• Are there not more comprehensive citations for the relationship between sleep and metabolic risk factors from systematic reviews and meta-analyses?

• What is meant by “increased dietary intake” (p.4, lines 17-18)?

• Although it is true that there have been few studies examining relationships between sleep and metabolic risk factors in younger samples, the authors fail to cite any relevant literature (e.g., Matthews et al, 2012; Javaheri et al, 2011).

• Please include citations and current guidelines re: sleep duration requirements across age groups (p.4, line 20).

• p.5, line 26 - there are other dimensions of sleep beyond duration and quality, I would not make this statement so categorical.

• The statements in p.5, lines 32-36 appear somewhat contradictory.

• I am somewhat confused by the statement “In order to develop appropriate recommendations among children and young adults” (p.5, lines 37-39) – your study did not include children below the age of 11, and the authors ultimately did not provide recommendations based on their results.

• p.5, lines 43-45 – citations needed for the statement that “device-measured sleep is more precise”? Furthermore, are you not downplaying the importance of your own study?

• The authors note that the present study focuses on sleep duration because it is “modifiable and relatively easy to record.” That statement needs a citation.

• Citation needed on p.5, lines 50-51 - metabolic syndrome/metabolic risk score is related to risk of what future health outcomes?

Matthews, et al. Sleep duration and insulin resistance in healthy black and white adolescents. Sleep. 2012;35(10):1353-58.

Javaheri, et al. Association of short and long sleep durations with insulin sensitivity in adolescents. J Pediatr. 2011;158:617–23.

Methods:

• Participants taking blood-pressure or lipid-lowering medications were excluded for analysis, which is good. Was data available on the % of participants who were taking medications for sleep purposes?

• Why were participants less than age 11 excluded (p.7, line 75), when the authors state that there is a need to “develop appropriate recommendations among children and young adults” (p.5, lines 37-39)?

• Would be helpful to know the % participants who were excluded due to age (<11 or >70 years), blood pressure or lipid-lowering medication use.

• What % of the sample were currently employed as shift workers?

Results:

• Table 1 – please provide raw data for individual metabolic risk factors, weekday sleep duration, and weekend sleep duration.

• The authors note that “the association appears strongest in the 36-50 age group” (p.12, line 203) – I would think that is better stated as the association was present only in the 36-50 age group, as no association exists in the other groups.

• Tables 2 and 3 – consider using a note to indicate which covariates were included in Models 1-3.

• The Results text and Table 2 uses phrases including “quadratic term”, “squared term,” and coefficient of age-adjusted sleep duration term2” – be consistent.

• p.13, line 213 – consider using the phrase “all ages” vs. “aggregated results” to be consistent with Table 2.

• The authors provide the p value on p.15, line 241 – would be helpful if they provided p values for other significant findings noted in the Results text.

Discussion

• The fact that the sample was largely white should be added as a limitation.

• Not sure what the authors mean by the sentence beginning “the recommendations change frequently with age…” (p.17, lines 294-295). Also provide citations.

• The authors found that the mid-age group appeared to be driving results for the effect of short sleep on metabolic risk but did not find significant results for the difference between weekday and weekend sleep in this age group. This is in contrast to evidence from Wong et al., who found that midlife adults who have greater social jetlag (difference in actigraphy-measured sleep between work and non-work days) showed a worse cardiometabolic profile.

• Provide citations for the sentences beginning “Screening for poor sleep…” and “There are a range of causes…” (p.20, lines 354-358).

• Provide age range for “young adults” (p.21, line 376).

• Need more explanation for the first sentence in the Concluding Remarks section – what, specifically, might be the clinical implications of heterogeneity between age groups?

Wong, et al. Social jetlag, chronotype, and cardiometabolic risk. J Clin Endocrinol Metab 2015;100:4612-20.

Reviewer #2: I think this will be a good addition to existing literature. Generalizability of the data may be difficult in this case since it was predominantly a white population, none the less the study was done thoughtfully.

Authors have highlighted most of the potential limitations which one may think have. The tables are well done. The technique and the stats are well done with no major limitations. I like the fact that it draws more spotlight on weekend and weekday sleep variation which is an area of future research.

6. PLOS authors have the option to publish the peer review history of their article (what does this mean?). If published, this will include your full peer review and any attached files.

Reviewer #1: No

Reviewer #2: No

---

## [Author Response · Author response to Decision Letter 0]

2 Nov 2020

Our response to reviewer's comments is uploaded as an attachment.

---

## [Decision Letter · Decision Letter 1]

11 Nov 2020

How do associations between sleep duration and metabolic health differ with age in the UK general population?

PONE-D-20-29508R1

Dear Dr. Winpenny,

We’re pleased to inform you that your manuscript has been judged scientifically suitable for publication and will be formally accepted for publication once it meets all outstanding technical requirements.

Kind regards,

Antonio Palazón-Bru, PhD

Academic Editor

PLOS ONE

Additional Editor Comments (optional):

Reviewers' comments:

Reviewer's Responses to Questions

**Comments to the Author**

1. If the authors have adequately addressed your comments raised in a previous round of review and you feel that this manuscript is now acceptable for publication, you may indicate that here to bypass the “Comments to the Author” section, enter your conflict of interest statement in the “Confidential to Editor” section, and submit your "Accept" recommendation.

Reviewer #1: All comments have been addressed

2. Is the manuscript technically sound, and do the data support the conclusions?

Reviewer #1: (No Response)

3. Has the statistical analysis been performed appropriately and rigorously? 

Reviewer #1: (No Response)

4. Have the authors made all data underlying the findings in their manuscript fully available?

Reviewer #1: (No Response)

5. Is the manuscript presented in an intelligible fashion and written in standard English?

Reviewer #1: (No Response)

6. Review Comments to the Author

Reviewer #1: (No Response)

7. PLOS authors have the option to publish the peer review history of their article (what does this mean?). If published, this will include your full peer review and any attached files.

Reviewer #1: No

---

## [Editor Report · Acceptance letter]

13 Nov 2020

PONE-D-20-29508R1 

How do associations between sleep duration and metabolic health differ with age in the UK general population? 

Dear Dr. Winpenny:

I'm pleased to inform you that your manuscript has been deemed suitable for publication in PLOS ONE. Congratulations! Your manuscript is now with our production department. 

Kind regards, 

on behalf of

Dr. Antonio Palazón-Bru 

Academic Editor

PLOS ONE